Multiple transgressions and slow evolution shape the phylogeographic pattern of the blind cave-dwelling shrimp Typhlocaris

http://orcid.org/0000-0002-6962-0262 Guy-Haim Tamar 1 2 tguy-haim@geomar.de
Simon-Blecher Noa 3
Frumkin Amos 4
Naaman Israel 4
Achituv Yair 3
1 Marine Ecology, GEOMAR, Helmholtz Centre for Ocean Research , Kiel , Germany
2 National Institute of Oceanography, Israel Oceanographic and Limnological Research , Haifa , Israel
3 The Mina and Everard Goodman Faculty of Life Sciences, Bar-Ilan University , Ramat Gan , Israel
4 Institute of Earth Science, The Hebrew University of Jerusalem , Jerusalem , Israel
Esteban María Ángeles
Electronic publication date: 2018 Jul 23
Publication date: 2018
Volume: 6
Electronic Location ID: e5268
Received 2018 Mar 4; Accepted 2018 Jun 29
Copyright: © 2018 Guy-Haim et al.
Copyright year: 2018
Copyright holder: Guy-Haim et al.
License: This is an open access article distributed under the terms of the Creative Commons Attribution License, which permits unrestricted use, distribution, reproduction and adaptation in any medium and for any purpose provided that it is properly attributed. For attribution, the original author(s), title, publication source (PeerJ) and either DOI or URL of the article must be cited.
License URL: https://creativecommons.org/licenses/by/4.0/

Keywords: Cave, Divergence time, Transgression, Mediterranean Sea, Stygofauna, Typhlocaris, Messinian Salinity Crisis, Subterranean, Vicariance, Speciation

Funding: The authors received no funding for this work.

==============================
Background

Aquatic subterranean species often exhibit disjunct distributions, with high level of endemism and small range, shaped by vicariance, limited dispersal, and evolutionary rates. We studied the disjunct biogeographic patterns of an endangered blind cave shrimp, Typhlocaris, and identified the geological and evolutionary processes that have shaped its divergence pattern.

Methods

We collected Typlocaris specimens of three species (T. galilea, T. ayyaloni, and T. salentina), originating from subterranean groundwater caves by the Mediterranean Sea, and used three mitochondrial genes (12S, 16S, cytochrome oxygnese subunit 1 (COI)) and four nuclear genes (18S, 28S, internal transcribed spacer, Histon 3) to infer their phylogenetic relationships. Using the radiometric dating of a geological formation (Bira) as a calibration node, we estimated the divergence times of the Typhlocaris species and the molecular evolution rates.

Results

The multi-locus ML/Bayesian trees of the concatenated seven gene sequences showed that T. salentina (Italy) and T. ayyaloni (Israel) are sister species, both sister to T. galilea (Israel). The divergence time of T. ayyaloni and T. salentina from T. galilea was 7.0 Ma based on Bira calibration. The divergence time of T. ayyaloni from T. salentina was 5.7 (4.4–6.9) Ma according to COI, and 5.8 (3.5–7.2) Ma according to 16S. The computed interspecific evolutionary rates were 0.0077 substitutions/Myr for COI, and 0.0046 substitutions/Myr for 16S.

Discussion

Two consecutive vicariant events have shaped the phylogeographic patterns of Typhlocaris species. First, T. galilea was tectonically isolated from its siblings in the Mediterranean Sea by the arching uplift of the central mountain range of Israel ca. seven Ma. Secondly, T. ayyaloni and T. salentina were stranded and separated by a marine transgression ca. six Ma, occurring just before the Messinian Salinity Crisis. Our estimated molecular evolution rates were in one order of magnitude lower than the rates of closely related crustaceans, as well as of other stygobiont species. We suggest that this slow evolution reflects the ecological conditions prevailing in the highly isolated subterranean water bodies inhabited by Typhlocaris.

Introduction

The biogeographic distribution patterns of populations of aquatic subterranean organisms (stygobionts) are characterized by a small range and high degree of endemism, originating from limited dispersal abilities and vicariant events, isolating the subterranean basins (Christman et al., 2005; Culver & Holsinger, 1992; Culver, Pipan & Schneider, 2009; Culver & Sket, 2000; Gibert & Deharveng, 2002; Porter, 2007). Sometimes the entire distribution of a stygobiont species is restricted to a single subterranean water body, exposing it to a substantial risk of extinction due to natural and anthropogenic pressures such as salt water intrusion, pollution, climate change, and overexploitation of groundwater for drinking and agricultural purposes, resulting in habitat destruction (Culver & Pipan, 2009; Danielopol et al., 2003; Gibert et al., 2009).

The aquatic subterranean fauna of the Levant is comprised of typical stygofauna (Por et al., 2013). Among them are at least four crustaceans, found in sites located along the Dead Sea Rift valley with congeneric taxa found in the Mediterranean coastal plain and even in brackish groundwater in the south of Israel. These obligate stygobionts are regarded as relicts of extinct marine fauna of ancient Mediterranean transgressions (Por, 1963). The most prominent members of this faunal assemblage are the large blind prawns of the genus Typhlocaris. Four species of this genus are known from four localities around the east Mediterranean Sea (Fig. 1). Each locality is inhabited by a different species with no congenerics in the open sea. Two species are known from Israel: T. galilea (Calman, 1909) from the Tabgha spring on Lake Kinneret shore, and the recently discovered T. ayyaloni (Tsurnamal, 2008), found in the karstic underground basin near Ramla, named Ayyalon cave, about 120 km south of Tabgha. The third species, T. salentina Caroli, 1923, was described from the Zinzulusa cave in Southern Italy and was recently found in other two caves in southern Italy (Froglia & Ungaro, 2001). The fourth species, T. lethaea Parisi, 1921, is known from Libya near Benghazi. In the IUCN Red List of Threatened Species, T. galilea and T. ayyaloni are defined as endangered, and T. salentina as vulnerable. No data later than 1960 on T. lethaea is available (De Grave, 2013).

Figure 1 Distribution map of Typhlocaris species (colored in red) based on spatial data from NatureServe and IUCN (International Union for Conservation of Nature).

The IUCN Red List of Threatened Species. Version 2014.1. (http://www.iucnredlist.org), downloaded on January 28, 2018. Map made using Natural Earth data (http://www.naturalearthdata.com).

The ancestor of Typhlocaris (“T. ancestor”) and the other marine taxa survived the regression of the Mediterranean Sea that occurred during the Messinian Salinity Crisis (MSC), 5.96–5.33 Ma, in caves and groundwater basins. Most probably, they were extirpated from the Mediterranean Sea waters when the Mediterranean desiccated and transformed to small hypersaline basins. During this crisis, the Mediterranean Sea lost almost all its Miocene tropical fauna, including those able to colonize subterranean waters (Por, 1975, 1986; Por & Dimentman, 2006). Therefore, the stranding of the Typhlocaris species and the separation from their common ancestor have likely preceded the MSC.

Two scenarios were proposed to explain the disjunct distribution of Typhlocaris (H1 and H2, Fig. 2). Por (1963, 1975) and Por & Dimentman (2006) suggested that Typhlocaris species have been stranded along the shores of a peri-Mediterranean Pliocene transgression, during the Zanclean (5.3–3.6 Ma). The timing of this scenario contradicts the pre-MSC stranding described above. According to Por (1963), the Typhlocaris species expanded their distribution into the Jordan valley when it was submerged for a brief period during the Zanclean marine transgression. The coastal plain was also submerged by this transgression that possibly also covered a part of the south of Israel (Por, 1963). Those faunal elements were left behind when the shore has retreated during the regression that followed the transgression in the early Pliocene. Similarly, Horowitz (2001) suggested that during the Pliocene, two successive transgressive cycles have occurred in the Zanclean and the Piacenzian, separated by a regression. Thus, according to this scenario, T. galilea and T. ayyaloni were separated together or at successive events from the Mediterranean fauna, and are thus sister taxa (H1, Fig. 2A).

Figure 2 Schemes describing the two hypotheses of development of the disjunct distribution of Typhlocaris.

(A) H1: the peri-Mediterranean transgression scenario. (B) H2: tectonic isolation of the eastern Galilee from the Mediterranean followed by stranding to the coastal aquifers by ingressions.

A recent study of the eastern Galilee (Rozenbaum et al., 2016) suggests a second scenario (H2, Fig. 2B). The marine transgression into the Dead Sea valley, bringing along T. galilea, was associated with a subsidence of the eastern Galilee. The Dead Sea rift valley, accommodating several water bodies, became tectonically isolated from the Mediterranean by the arching uplift of the central mountain range of Israel. This uplift also divided the groundwater basins of the Dead Sea basin from those associated with the Mediterranean, thus resulting in an earlier divergence of T. galilea than the divergence of its sister species. In contrast, the other three Typhlocaris species were found in coastal to inland aquifers that are not isolated from the Mediterranean by a tectonic barrier. They could be stranded in the coastal aquifers by a regression that was not necessarily associated with a tectonic event. This scenario (H2) is supported by the finding of marine macrofossils within the late Miocene Bira Formation of the SE Galilee–Jordan valley indicating its association with a marine transgression (Shaked-Gelband et al., 2014). Ar–Ar dates of volcanics interbedded within the Bira Formation show that the earliest marine invasion into the SE Galilee–Jordan valley happened between 11 and 10 Ma (Rozenbaum et al., 2016; for earlier dating see Shaliv, 1989). Ongoing subsidence of the SE Galilee basin, coupled with rising sea level, resulted in the invasion of the Mediterranean water and establishment of a seaway that connected it to the evolving Dead Sea Rift in the east, as represented by parts of the Bira Formation. Seawater could have flowed to the SE Galilee basin either due to global sea level rise above the low barrier near the coastline or due to tectonic subsidence of the Yizre’el valley which had already started to develop. The detachment of this region from the Mediterranean occurred ca. seven Ma, when the Mediterranean Sea level started falling during the Messinian, followed by freshwaters gradually replacing the saline waters of the Bira lagoon. Thus, the main marine ingression is constrained to the Tortonian, prior to the MSC. Further to the NE, within the Hula valley, Syria and Lebanon, there is no indication of this marine transgression, demonstrating that the marine water came from the Mediterranean and not from the NE (Rozenbaum et al., 2016). This is consistent with the circum-Mediterranean distribution of the four Typhlocaris species.

The main objectives of our study were: (1) to reveal the phylogenetic relationships of the Typhlocaris species, and to use these patterns to (2) infer the geological and evolutionary processes that have shaped their divergence patterns.

Materials and Methods

Species sampling, genes, and outgroup selection

Specimens of T. galilea were collected by us, in the covered pool collecting the water of Tabgha spring (32°52′20″N 35°33′00″E) on Lake Kinneret shore (NPA permit 37920). T. ayyaloni was collected from the underground groundwater pond in Ayyalon cave (31°54′37″N 34°55′39″E), two specimens of T. salentina were provided by Dr. G. Messana Firenze—Italy from two caves in the vicinity of Bari, Italy, Lu Bissu cave (39°59′42″N 15°57′58″E) and Mola di Bari cave (41°03′36″N 17°05′24″E). All samples were fixed and stored in 95% ethanol at −20 °C until DNA extraction. The locality of the fourth species, T. lethaea, is restricted to Lete Cave, near Benghazi, Libya, and is not accessible. The two specimens of T. lethaea, collected by Parisi a century ago (1921), and stored in the Museum National d’Histoire Naturelle, Paris, did not yield DNA.

DNA extraction, amplification, and sequencing

DNA was extracted using Macherey–Nagel genomic DNA isolation kit (Düren, Germany), following the manufacturer’s recommended protocol. The primers used for gene amplification are detailed in the Supplemental Information, including both primers from former studies and newly designed primers for this study (Table S1). REDTaq ReadyMix R2523 (Sigma–Aldrich, St. Louis, MO, USA) was used for sequence amplification by PCR (Saiki et al., 1988). Amplification was carried out in a personal combi-thermocycler (Biometra, Germany) according to the profiles listed in Table S1. PCR products were purified by centrifugation using a High Pure PCR product purification kit (Roche Diagnostics GmbH, Mannheim, Germany) or by Mclab laboratories (San Francisco, CA, USA). PCR products were sequenced on both strands using an ABI PRISM 3100 Genetic Analyzer (Applied Biosystems, Foster City, CA, USA) by McLab laboratories (San Francisco, CA, USA).

Three mitochondrial genes (12S rRNA; 16S rRNA; cytochrome oxygnese subunit 1 (COI)) and four nuclear genes (18S rRNA; 28S rRNA, internal transcribed spacer (ITS); Histon 3 (H3)) were chosen for analysis. For phylogenetic inference of all seven gene partitions, we used Ephyrina figueirai Crosnier & Forest, 1973 (family: Acanthephyridae), and Palaemon elegans Rathke, 1837 (family: Palaemonidae), as outgroup species that belong to the Caridea, the same infraorder of Typhlocaris, because sequences of the seven genes used in our analysis were available in GenBank. Considering that both Palaemon and Typhlocaris belong to the same superfamily (Palaemonoidea), and since Palaemonoidea is paraphyletic (Kou et al., 2013), E. figueirai was chosen as a root node.

The Typhlocaris sequences were deposited in GenBank under accession numbers KY593415–KY593454. In addition to the newly generated sequences, two sequences of T. salentina were obtained from GenBank and included in the molecular analysis. The list of taxa, localities and GenBank accession numbers included in the analysis is detailed in Table S2.

Phylogenetic analyses

Sequence alignment was conducted using ClustalX embedded in MEGA v6.0 (Tamura et al., 2013).The sequences were concatenated to form a multi-gene matrix using Geneious v7.1 (http://www.geneious.com/), including the three Typhlocaris sequences and two outgroups, delimited into seven partitions, one for each gene. MEGA v6.0 (Tamura et al., 2013) was used in order to select the best fitting substitution model for each partition according to the Bayesian Information Criterion (Table 1).

Table 1 Nucleotide analysis and substitution models selected (out of 24 candidate models) for all the genes/partitions.

Partition	Length (bp)	Informative positions	Variable positions	Model	Nst-rates	
12S	394	161	236	T92+G	6—Gamma	
16S	972	160	221	HKY+G	2—Gamma	
COI	663	254	286	GTR+G+I	6—Gamma	
18S	1914	263	342	K2+G	2—Gamma	
28S	2059	306	659	T92+G	6—Gamma	
ITS	1795	612	1523	T92+G	6—Gamma	
H3	358	50	97	K2+G	2—Gamma	

Maximum-likelihood (ML) analysis of the aligned partitions was conducted using RAxML v8.2.9 (Stamatakis, 2014) on XSEDE server in the CIPRES Science Gateway portal (https://www.phylo.org/portal2) using a GTRCAT model of evolution with 50 rate categories with 1,000 bootstrapping replicates. Bayesian Metropolis coupled Markov chain Monte Carlo (B-MCMC) analyses were conducted with MrBayes v3.2 on XSEDE with GTR model (Ronquist et al., 2012). Search was conducted with four chains (three cold, one hot) with trees sampled every 100 generations. Three 100 generations analyses were conducted to verify likelihood convergence and burn-in parameter.

Estimation of evolutionary rates

Since the molecular clock calculations for cave-dwelling species are often contentious (Page, Humphreys & Hughes, 2008), we used multiple genes and a relaxed molecular clock approach (Drummond et al., 2006). To estimate the divergence time of Typhlocaris species, we first performed analyses based on accepted molecular evolution rates of the mitochondrial genes COI and 16S rRNA for crustaceans: 0.0140 nucleotide substitutions per Myr (Knowlton & Weigt, 1998), and 0.0090 substitutions per Myr (Sturmbauer, Levinton & Christy, 1996). As an alternative approach, we used a calibrated tree based on a regional geological event. A similar approach was applied by Bauzà-Ribot et al. (2012) that used two paleogeographic events as a calibration point to establish the divergence pattern of the stygobiont family Metacrangonyctidae (Amphipoda). The top of Bira formation, dated to seven Ma (Rozenbaum et al., 2016), marks the end of the marine connection between the Mediterranean and the Dead Sea valley. Therefore we assume that this event indicates the isolation of T. galilea from its sister taxa, and we used it as a calibration node. Using Bira formation as a calibration node, solely allowed the estimation of the divergence time of the sister species, T. ayyaloni and T. salentina, and thus infer the geological event that led to this separation.

Bayesian evolutionary analysis was used to obtain the evolutionary rates of COI and 16S genes under the favored tree topology, based on the ML analysis. A relaxed-clock MCMC approach using the uncorrelated log-normal model was implemented in BEAST v2.4 (Drummond & Bouckaert, 2015) on XSEDE server in the CIPRES Science Gateway portal (https://www.phylo.org/portal2/), using 10 million generations, and sampling every 1,000th generation. Models of sequence evolution for each gene were determined using the corrected Akaike information criterion in JModelTest v2.1 (Darriba & Posada, 2014, Table 2) on XSEDE server. The Yule process was chosen as speciation process for both genes. Log files were analyzed with Tracer v1.6 (Rambaut et al., 2015), to assess convergence and confirm that the combined effective sample sizes for all parameters were larger than 200, in order to ensure that the MCMC chain had run long enough to get a valid estimate of the parameters (Drummond & Rambaut, 2007). All resulting trees were then combined with LogCombiner v1.8.2, with a burn-in of 25%. A maximum credibility tree was then produced using TreeAnnotator v2.1.2 (Rambaut & Drummond, 2015).

Table 2 Divergence times (and 95% CI) for Typhlocaris species as estimated using Bayesian evolutionary analysis method calculated using COI and 16S gene molecular evolution rates and using calibration based on Bira formation.

Clade divergence	Gene	Node age (Myr) (range) non-calibrated	Calibration node	Node age (Myr) (range) calibrated	Posterior probability	
Typhlocaris	COI	13.4 (10.6–14.0)	–	19.9 (17.3–22.5)	0.48	
16S	19.1 (16.5–22.2)	41.5 (35.8–48.5)	1.00	
(T. ayyaloni + T. salentina)—T. galilea	COI	3.7 (3.0–4.5)	7.0 (Bira)		1.00	
16S	3.3 (2.3–4.2)		1.00	
T. ayyaloni–T. salentina	COI	3.2 (2.4–3.8)	–	5.7 (4.4–6.9)	0.76	
16S	2.6 (1.6–3.4)	5.8 (3.5–7.2)	0.76	

Results

The concatenated alignment of the seven genes was 7,761 bp long, out of which 1,645 were parsimonious informative. The substitution models selected for all the genes/partitions with the corrected Akaike Information Criterion and the Bayesian Information Criterion scores is presented in Table 1. Figure 3 presents a ML tree of the concatenated sequences, showing that T. salentina and T. ayyaloni are more closely related to each other than either of them is to T. galilea. Out of the seven genes used for the analysis, five gene sequences (ITS, 28S, COI, 12S, 16S) presented this topology. The remaining gene trees, of 18S and H3, had slightly different topology. However, the bootstrap support of the nodes connecting Typhlocaris species in these two trees was less than 50%. The topology of the five gene phylogenetic tree supports the hypothesis suggesting that T. galilea was separated from its presumed marine ancestor earlier than the separation of T. ayyaloni and T. salentina (H2, Fig. 2).

Figure 3 Multi-locus maximum-likelihood tree of the genus Typhlocaris, based on combined 12S rRNA + 16S rRNA + COI + 18S rRNA + 28S rRNA + ITS + H3 genes (total 7,761 bp).

At each node, the number above the branch indicates the percentage of ML bootstrap support (1,000 replicates) from RAxML analysis with the GTRCAT model of evolution. The number below the branch at each node indicates the Bayesian posterior probability expressed as a decimal fraction for nodes that received at least 50% support in at least one analysis. The scale bar denotes the estimated number of nucleotide substitutions per site.

Using the common evolutionary rates for crustacean COI and 16S genes, 0.0140 and 0.0090 substitutions/Myr, respectively (Knowlton & Weigt, 1998; Sturmbauer, Levinton & Christy, 1996), the divergence time estimations for T. galilea and T. salentina–T. ayyaloni clade were 3.7 (3.0–4.5) and 3.3 (2.3–4.2) Ma, respectively (means (95% highest probability density intervals)). The divergence time between T. ayyaloni and T. salentina was estimated as 3.2 (2.4–3.8) Ma according to COI and as 2.6 (1.6–3.4) according to 16S (Table 2). These estimations suggest that the divergence of Typhlocaris species has happened two million years after the Zanclean reflooding of the Mediterranean Sea, thus under no apparent vicariant conditions.

Using seven Ma as the detachment time that isolated T. galilea from the Mediterranean Sea (top Bira formation), the divergence time of T. ayyaloni and T. salentina was according to COI gene—6.0 (4.5–7.2) Ma and according to the 16S gene—5.9 (3.6–7.4) Ma (Table 2), suggesting that these are relicts of the last high level of the Mediterranean Sea before the MSC. The computed evolutionary rates for COI—0.0077 substitutions/Myr and for 16S—0.0046 substitutions/Myr, are notably lower than the molecular clock rates found in previous crustacean studies (Table 3). The evolutionary rates of ITS, 28S, and 12S were 0.0104, 0.0184, 0.0115 substitutions/Myr, respectively.

Table 3 Comparison between the COI and 16S molecular evolution rates estimated in this and previous crustacean studies.

Gene	Stygofauna	Non-Stygofauna	
Species	Substitutions/Myr	Species	Substitutions/Myr	
COI mtRNA	Typhlocaris spp.[1]	0.0077	Alpheus spp.[2]	0.0140	
Stygiocaris spp.[3]	0.0133–0.0516	Halocaridina spp.[7]	0.2000	
Stenasellus spp.[6]	0.0125			
16S rRNA	Typhlocaris spp.[1]	0.0046	Sesarma spp.[4]	0.0065	
Stygiocaris spp.[3]	0.0055–0.0103	Uca spp.[5]	0.0090	
Notes:

[1] This study.

[2] Knowlton & Weigt (1998).

[3] Page, Humphreys & Hughes (2008).

[4] Schubart, Diesel & Hedges (1998).

[5] Sturmbauer, Levinton & Christy (1996).

[6] Ketmaier, Argano & Caccone (2003).

[7] Craft et al. (2008).

Discussion

Marine regressions are the most significant vicariant events forming physical barriers and structuring stygoboint speciation (Boutin & Coineau, 2012; Culver, Pipan & Schneider, 2009; Notenboom, 1991; Porter, 2007; Stock, 1993). Other influential vicariant events include uplilft of mountain ridges (Bauzà-Ribot et al., 2012; Humphreys & Danielopol, 2005; Reid et al., 2002), and events that destroy or close off aquatic dispersal corridors (Barr & Holsinger, 1985; Holsinger, 2012). Using molecular techniques, we established the phylogeny of Typhlocaris species, and showed that T. salentina (Italy) and T. ayyaloni (Israel) are sister species, both sister to T. galilea (Israel). These phylogeographic relationships indicated that more than one vicariant event have shaped the speciation pattern of Typhlocaris. First, T. galilea was tectonically isolated from the Mediterranean Sea by the arching uplift of the central mountain range of Israel, ∼seven Ma (Matmon, Wdowinski & Hall, 2003; Wdowinski & Zilberman, 1997). Later, T. ayyaloni and T. salentina were stranded and separated by a marine regression ∼six Ma, as a result of the MSC.

The fourth Typhlocaris species, T. lethaea, was missing from our analysis due to the inaccessibility of Lete Cave, Libya, where it is found. Hypothetically, adding T. lethaea to the phylogenetic analysis, could have resulted in a modified tree topology, and potentially, in different scenario of speciation (e.g., finding that T. galilea and T. lethaea are sister species will compel a modification of the inferred speciation model). The long branch of one of the T. salentina specimens likely reflects the difference between populations originating in different cave systems in southern Italy, where the samples were collected (Lu Bissu and Mola di Bari caves). Both the effect of T. lethaea on the phylogeographic pattern of Typhlocaris, and the population genetics of T. salentina in the caves and wells of Salento and southern Murge karst systems, warrant each for an independent study.

Commonly, the final closure of the Isthmus of Panama that has occurred approximately three Ma (Coates et al., 1992; Keigwin, 1978, 1982; O’Dea et al., 2016) is used for estimation and calibration of divergence time of crustaceans. Knowlton & Weigt (1998) and Williams et al. (2001) found that the substitution rate of COI is 0.0140 per Myr. This finding is based on the pairs of transisthmian snapping shrimp Alpheus from Panama: A. estuarensis—A. colombiensis, and A. antepaenultimus—A. chacei. Schubart, Diesel & Hedges (1998) calibrated the substitution rate of 16S rDNA using trans-isthmian pairs of crabs of the genus Sesarma (Grapsidae) and then used this rate to estimate a date for the origin of the Jamaican lineage Sesarma, the substitution rate of Sesarma was 0.0065 per Myr. Sturmbauer, Levinton & Christy (1996) used the same gene from populations of the fiddler crab Uca vocator, from either side of the Isthmus of Panama to estimate divergences rates of Uca. The sequence divergence rate was 0.0090 per Myr; this rate was used to estimate the time divergence between clades of terrestrial Uca from different parts of the globe.

Craft et al. (2008) and Page, Humphreys & Hughes (2008) that studied the phylogeography of atyids did not use the rates of transisthmian organisms to calibrate the molecular clock but estimated it independently for the studied taxa. Craft et al. (2008) studied Halocaridina from the Hawaiian Archipelago. To calibrate the molecular clock, they used the age of the earliest eruption of Kilauea volcano in Hawaii, 50–100 Ka, and the genetic data of the groups of Halocaridina that occur along the flank of this volcano. They found an exceptionally high divergence rate of 0.2 per Myr in COI gene of Halocaridina. They noted that this rate is in sharp contrast to the commonly utilized evolution rates for arthropods 0.0140–0.0170 per Myr (Williams et al., 2001). Page, Humphreys & Hughes (2008) studied the cave atyids Stygiocaris from Cape Range area in Western Australia. It is accepted that the emergence of the Cape Range Anticline in the Miocene isolated Stygiocaris lancifera and S. stylifera, leading to their speciation, therefore, Page, Humphreys & Hughes (2008) used this event, 7–10 Ma, as a calibration point to estimate rates of molecular divergence. This yielded a wide range of evolutionary rates for the S. lancifera/stylifera node: 0.0133–0.0516 substitutions/Myr in COI and 0.0055–0.0103 substitutions/Myr in 16S, relatively lower than other atyid studies, but still higher than the rate we found for Typhlocaris.

Our estimated low evolutionary rates in Typhlocaris correspond with the analysis of Zakšek et al. (2009) that studied the phylogeography the cave shrimp Troglocaris anophthalmus. To estimate the divergence time they referred to the divergence rate of COI used for transisthmian species of Alpheus across the Isthmus of Panama (Knowlton & Weigt, 1998). Zakšek et al. (2009), therefore, stated that for Troglocaris, the rate calculated by Knowlton & Weigt (1998) can be used only for estimation of the order of magnitude of divergence time because it is the most commonly used rate for decapods. Nonetheless, they found COI patristic distances between phylogroups that are much lower (0.05–0.08) than the accepted patristic COI distance of 0.16 substitutions per nucleotide position found to optimally separate intra-from interspecies divergence in other crustaceans (Lefébure et al., 2007).

The rates found by us are in one order of magnitude lower than those found for Alpheus, the common crustacean used for calibration of divergence time (Knowlton & Weigt, 1998), and lower or similar to the rates of other stygobionts (Table 3). An exception is the case of the stygobiont amphipod family Metacrangonyctidae, which was shown to undergo rapid evolution using mitochondrial protein-coding genes (Bauzà-Ribot et al., 2012). The average rate estimated by Bauzà-Ribot et al. (2012) was 0.1090 substitutions/Myr, one order of magnitude higher than the rates acceptable for other crustaceans. They suggested that this high rate might result from frequent population bottlenecks. Evolutionary rates, even of the same gene, may vary between different genera within the same order—indicating that evolutionary rates are not related only to the taxonomic position but also, or mainly, to ecological conditions. We therefore did not use the previously reported substitution rate but the known geological data of the area where Typhlocaris occurs to infer its divergence rate and time. The lower divergence rates found for Typhlocaris compared with other crustaceans lead us to the suggestion that the low rates are related to the ecological conditions of the Typhlocaris habitat. Typhlocaris and other stygobionts are found in isolated subterranean basins where species diversity is very low, relative to the regional diversity (Gibert et al., 2009), reducing interspecific competition. The environmental factors in these habitats are stable, lacking fluctuations. Predators are typically missing in subterranean habitats, resulting in truncated food webs (Gibert & Deharveng, 2002). Additionally, evolution rates were correlated with metabolic rates (Martin & Palumbi, 1993). Species with low metabolic rates (e.g., deep-sea fauna) are generally characterized by reduced nucleotide substitution rates. It was hypothesized that limited light reduces visual predation pressure and selects for reduced locomotory ability and metabolic capacity (Da Silva et al., 2011). This may be just as well the case of stygobiont evolution. Thus, the combined unique ecological and biological conditions (dark habitat, environmental stability, low richness, lack of interspecific competition) may lead to stability and low rate of gene divergence. This is in agreement with the statement of Mayr (1963) that competition and allopatry are important elements of speciation and evolutionary divergence.

Culver (1976) noted that the most striking feature of the organization of Appalachian cave—stream communities is the reduction in intensity of competition. One of the suggested explanations is that, with increasing time in caves, species evolve a life-history strategy of high metabolic efficiency and low reproductive rate, a strategy that may itself reduce interspecific competition. We thus may assume that the higher divergence rates found in non-stygobiont crustaceans are related to competition. The classical taxa used for calibration of molecular dating are the 18 species of Alpheus at both sides of the Isthmus of Panama (Knowlton & Weigt, 1998). Knowlton (1993) observed aggressive behavior among species including individuals that belong to a nominal species from both sides of the Isthmus of Panama, supporting our assumption on the role of competition in delimiting evolutionary rates.

Using evolutionary biology, we can identify processes that promote or maintain phenotypic and genetic diversity in natural populations. This is of a great importance, particularly when the studied organisms are under high risk of becoming extinct. While many studies confirmed that interspecific competition and environmental variation drive genetic diversification, there is little phylogeographic evidence linking environmental stability with low genetic variation. Further molecular investigations of stygobionts and other organisms of stable environments will shed light on universality of their temporal mode of speciation.

Conclusion

Our results indicated that two separate vicariant event shaped the distribution patterns of the blind cave-dwelling shrimp Typhlocaris. During the late Miocene, T. galilea was tectonically isolated from the Mediterranean Sea by the arching uplift of the central mountain range of Israel, ca. seven Ma. During the MSC, T. ayyaloni, geographically adjacent to T. galilea, and T. salentina were stranded and separated by a marine transgression. A future investigation of the divergence time of T. lethaea may shed more light on the transgression events leading to the disjunct phylogeographic pattern of Typhlocaris. Furthermore, the evolutionary rates of Typhlocaris estimated in this study (0.0077 substitutions/Myr in COI and 0.0046 substitutions/Myr in 16S rRNA) were in one order of magnitude lower than the rates of closely related crustaceans, and lower than other stygobiont species. These low rates may result from the low predation stress and the low diversity, leading to low interspecific competition, which characterizes the highly isolated subterranean habitats inhabited by Typhlocaris.

Supplemental Information

Supplemental Information 1 Table S1. List of the primers used for gene amplification in this study and PCR profiles.

Click here for additional data file.

Supplemental Information 2 Table S2. GenBank accession numbers of Typhlocaris.

Click here for additional data file.

We thank Dr. G. Messana Firenze of Instituto per lo Studio degli Ecosistemi, Florence, Italy, for providing specimens of Typhlocaris salentina from Lu Bissu cave and Mola di Bari cave. The Museum National d’Histoire Naturelle in Paris for the use of T. lethaea specimens. Dr. Hanan Dimentman for assisting with the study of T. ayyaloni. Francisco R. Barboza and Markus Franz for helping with map preparation. We thank the anonymous reviewers for their valuable comments and in helping to improve the manuscript. This article is dedicated to the memory of Prof. Francisc D. Por, who initiated the study of relict aquatic fauna of the Jordan rift valley.

Additional Information and Declarations

Competing Interests

Author Contributions

Field Study Permissions

Data Availability

The authors declare that they have no competing interests.

Tamar Guy-Haim conceived and designed the experiments, performed the experiments, analyzed the data, prepared figures and/or tables, authored or reviewed drafts of the paper, approved the final draft.

Noa Simon-Blecher conceived and designed the experiments, performed the experiments, approved the final draft.

Amos Frumkin contributed reagents/materials/analysis tools, authored or reviewed drafts of the paper, approved the final draft.

Israel Naaman contributed reagents/materials/analysis tools, approved the final draft.

Yair Achituv conceived and designed the experiments, performed the experiments, analyzed the data, prepared figures and/or tables, authored or reviewed drafts of the paper, approved the final draft.

The following information was supplied relating to field study approvals (i.e., approving body and any reference numbers):

Field experiments were approved by Israel Nature and Parks Authority (NPA permit 37920).

The following information was supplied regarding data availability:

The authors confirm that all data underlying the findings are fully available without restriction. All DNA sequences generated in this research were deposited in GenBank. The list of primers used and designed for this study and the list of taxa, localities and GenBank accession numbers are detailed in Tables S1 and S2, respectively.

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
