# Peer review of "Multiple transgressions and slow evolution shape the phylogeographic pattern of the blind cave-dwelling shrimp Typhlocaris"

_PeerJ, doi:10.7717/peerj.5268_

## Round 0.1 · original submission · Major Revisions

Your manuscript has to be revised, mainly in those aspects regarding methodology and evolution.

Reviewer 1 ·

Basic reporting

The paper reports molecular data of three of the four species of the genus Typhlocaris, which allows to discriminate between two alternative scenarios for their origin. These are rare species with high conservation interest, but also very interesting from an evolutionary and biogeographic perspective. I think the data and the general conclusions re. the origin of the species are valid, but there are some methodological aspects, and some considerations on the evolutionary rates, that should be revised.

Some detailed comments follow:

Abstract (also similar expresions in the text) “T. salentina (Italy) and T. ayyaloni (Israel) are more closely related than T. galilea (Israel).” The text is confusing, possible alternatives are: "more closely related to each other than either of them is to T. galilea", or better, "they are sister species, and both sister to T. galilea."

line 62: “is the large blind prawn of the genus Typhlocaris.” Should be plural.
line 135: give commercial brand and city.
lines 150-151: the two “transisthmian” pairs are used only for comparison in the Discussion, there is no need to specify this in the Methods.
lines 195-196: there is no need to mention the results of a NJ tree. This is not a phylogenetic method.

In Table 1, the informative positions as with or without outgroups? Also, the BIC and AICs of the best models on their own are meaningless: they should be compared with the values obtained under different models. Unless all the values obtained with the different models tested are given, so that they can be compared, there is no point in giving those of the best models.

Experimental design

lines 148-149: more details are needed re. the choice of outgroups. There are only two species, it is said that both belong to the infraorder Caridae, but no family or superfamily is given. According to current taxonomy, it seems that one of them belongs to the same superfamily as Typhlocaris (Palaemon), but the other to a different superfamily (Ephyrina). If so, the trees should be rooted in Ephyrina, not in the split between Typhlocaris and (Palaemon+ Ephyrina) (see e.g. a recent molecular phylogeny of Palaemonoidea in Kou et al., 2013, Inv. Syst.), as this renders Palaemonoidea paraphyletic. This has no implication for the topology of the genus Typhlocaris, but could have some implications for the dating.

In reference to this question, did the author build a calibrated, ultrametric tree? From their explanations is not clear if they did, but they certainly should, either with MrBayes, BEAST or other similar method. They should also figure it – and make sure that the outgroups are rooted properly.

Line 170++: The tree is calibrated with the split between T. galilea and its putative sister. This node is thus fixed a priori, and should not be considered a result, or given the impression that is a finding of the work. In some cases this is not clear, e.g. in lines 212-216. It should be more clear that the dating of this node is not a result of the study. This means that the evolutionary rates are fixed by this a-priori dating, and should be considered with caution.

Validity of the findings

Lines 265++ The whole discussion on the reasons for the low evolutionary rates is just far too speculative, especially considering that, as noted above, they are the result of an a-priori calibration of a node. The easiest solution to the problem is to change the age of the node: perhaps the separation between T. galilea and its sisters was not 7Ma, but in a more recent time. Then the problem of the very slow rates would be solved without the need to invoke obscure reasons (see below).

Lines 265-267: I cannot follow this argument. The authors say that as evolutionary rates differ between genera within the same order, differences should be mainly due to ecological differences between them. There are many other potential sources for these differences: effective population size, generation time, body size, metabolic rate, presence of selective pressures….. They could be related to ecological differences, but they may not.

Lines 280-283, 288-289,311-314: same as above. The authors do not give any convincing reason for which the subterranean environment, or the lack of competition, should slow down the evolutionary rate. In fact, there are many subterranean groups, even among crustaceans, that do not show any slow down in evolutionary rate, despite having a similar ecology and way of life that then species of Typhlocaris (see e.g. Jurado-Rivera et al., 2017 Sci. Rep., or Bauza-Ribot et al., 2012 Curr. Biol., not to mention many other papers in other arthropod lineages).

Additional comments

A nice work, which conclusions I think are well founded, but that should be presented more convincingly and without needless speculation.

Reviewer 2 ·

Basic reporting

The manuscript is written in English and easy to follow, literature well referenced and relevant, figures of high quality and raw data are available. The reporting, however, should be more focused. In the Introduction authors formulate two biogeographic questions. The two questions are not explicitly linked with the methods, and major part of the Discussion does not discuss the two question. Authors should try to make the manuscript more coherent, the Discussion could be shortened for at least for 50%.

Experimental design

The research is within the scope of the journal, the methods are appropriate for addressing the question and sufficiently well described. However, I have serious problems with experimental design. The objectives of the MS are to 1) reveal phylogenetic relationship of the Typhlocaris species and 2) to infer geological processes that have shaped their divergence pattern. The text that follows is only partly following the objectives.

The authors studied the relationship of three out of four species of this genus. This part is mostly clear. An exception is the long-branch among Italian populations, which may indicate an error, and authors should discuss this result. Furthermore, the authors studied only three species. I understand it is not possible to get the fourth species, but make this impediment clear and clarify it in the Introduction / where you postulate the hypotheses. Please note that it may impact the second aim!

The second aim is less clear to follow. Authors are not explicit, what test they used for testing hypotheses. On the one hand, Figure 2 implies that phylogenetic hierarchy itself is a test for either of hypotheses. I am fine with this, once the putative problem with the missing fourth species is accommodated. On the other hand, authors devoted much debate to timing and give an impression that they searched for temporal congruence in geological and splitting events. In theory, this approach would strengthen the test, but not in the way the authors did. If I understood the logic correctly, the described approach leads to cyclic reasoning. The problem is that they used one of the studied events as a calibration point, and then from calibrated phylogeny inferred the concordance between branching events and geological events. This impression is further strengthened by extensive discussion on molecular clock, whereas the paleogeographic and paleogeologic events are barely mentioned. The authors can overcome this issue in two ways. First, they can calibrate their phylogeny using some external calibration point (a calibration point outside the studied clade) and thereby assure that the studied events are at least seemingly independent from a priori assumptions used in testing. Alternatively, authors can use any of proposed molecular clocks, and check whether it fits to the data.

Finally, I recommend that authors stick to the questions. They should adjust the methods in order to assure the question is properly address, and in Discussion they should stick to the questions, rather to the methodological issue. I acknowledge that the latter is relevant, but it does not fall into the scope of the paper. The Discussion can be shortened, but I miss e.g. comments on missing species (which may influence the phylogenetic structure), or speciation that took place in the sea, prior the transgression events.

Validity of the findings

I suggest that the analyses are repeated as suggested above and the findings re-assessed.

Additional comments

I have read the MS, a report is appended below, some comments are embedded directly into attached PDF. Overall, I think the MS unveils some new insights into phylogenetic relationships of the cave shrimps and should be published subject to revision.

Annotated reviews are not available for download in order to protect the identity of reviewers who chose to remain anonymous.

---

## Round 0.2 · Minor Revisions

Your manuscript has been improved but it still needs some modifications. Please, follow the recommendations given regarding better focus on the the aim of the paper (see 'comment for the author' below)

Reviewer 1 ·

Basic reporting

The authors have revised the manuscript following the indications of the referees, and most of the objections have now been addressed appropriately. There are, however, some issues remaining, but mostly referring to the way results are presented and discussed, and not substantial.

Experimental design

no comment.

Validity of the findings

no comment.

Additional comments

I still find the line of argument of the paper a bit confusing, or at least not very clearly exposed. As I see it, the authors start with two different hypotheses for the origin of the studied taxa, which can be separated by the topology of the tree alone. Their tree supports unambiguously one of the topologies, favouring the pre-Messinian origin of the divergences within the genus Typhlocharis, so that with this the issue of the biogeographic origin is considered to be settled. The purpose of the use of BEAST is thus only to obtain the evolutionary rates of the genes studied under the favoured topology, and compare them with the rates given for Crustacea in other published works - it is not relevant to establish the biogeography of the genus, as the authors already accept its pre-Messinian origin based on the topology alone. In consequence, they dismiss the dating using the published rates and accept those obtained with the pre-Messinian calibration point without further discussion. If my interpretation is correct, this should be made more explicit through the paper.

And I am still thinking that the link between slow evolutionary rates and "ecology", and more particularly competition, is purely speculative and the authors do not provide any hard data to support it. The Discussion may be open to some speculation, but I would recommend the authors to be more cautious when interpreting their results.

---

## Round 0.3 · accepted · Accept

After reading the rebuttal letter and see the revised version of the manuscript, I am pleased to confirm that your paper has been accepted for publication. Your manuscript was improving according to the reviewers suggestions,

Thank you for submitting your work to this journal.

#